

# Exploring randomness in autism

Vasileios Mantas[1], Vasileia Kotoula[2] and Artemios Pehlivanidis[1]

[1] 1st Department of Psychiatry, Aiginiteion Hospital, National and Kapodistrian University of Athens, Athens, Attica, Greece
[2] Laskaridis Foudation, Athens, Attica, Greece

## ABSTRACT

**Introduction**. The fast, intuitive and autonomous system 1 along with the slow, analytical and more logical system 2 constitute the dual system processing model of decision making. Whether acting independently or influencing each other both systems would, to an extent, rely on randomness in order to reach a decision. The role of randomness, however, would be more pronounced when arbitrary choices need to be made, typically engaging system 1. The present exploratory study aims to capture the expression of a possible innate randomness mechanism, as proposed by the authors, by trying to isolate system 1 and examine arbitrary decision making in autistic participants with high functioning Autism Spectrum Disorders (ASD).

**Methods**. Autistic participants withhigh functioning ASD and an age and gender matched comparison group performed the random number generation task. The task was modified to limit the contribution of working memory and allow any innate randomness mechanisms expressed through system 1, to emerge.

**Results**. Utilizing a standard analyses approach, the random number sequences produced by autistic individuals and the comparison group did not differ in their randomness characteristics. No significant differences were identified when the sequences were examined using a moving window approach. When machine learning was used, random sequences' features could discriminate the groups with relatively high accuracy.

**Conclusions**. Our findings indicate the possibility that individual patterns during random sequence production could be consistent enough between groups to allow for an accurate discrimination between the autistic and the comparison group. In order to draw firm conclusions around innate randomness and further validate our experiment, our findings need to be replicated in a bigger sample.

Corresponding author
Vasileios Mantas,
mantas.vasilios@yahoo.com

## INTRODUCTION

One well established model of thinking in cognitive psychology proposes a distinction of the decision making mechanisms in the fast, intuitive and autonomic system 1 and the slow, analytical and more logical system 2 (*Kahneman, 2003*). Activities such as detecting that one object is more distant than another or performing simple mathematical calculations are usually attributed to system 1 while other behaviours where attention and cognitive processes are actively employed, like filling out a tax form, are attributed to system 2 (*Evans & Stanovich, 2013*; *Kahneman, 2003*; *Wason & Evans, 1974*). The two systems can operate independently but they can also interact and influence one another. Emotions, memories

and impulses generated by system 1 can inform system 2 which, in its turn, can silence, guide or train system 1 (*Evans & Stanovich, 2013*).

In this paper and based on the dual-system model, we will briefly analyse the role of these systems in arbitrary decision making. We will then talk about the concept of randomness and how we believe that would come into play when such decisions are attempted and explain why we are interested in arbitrary choices in autism. Finally, we will talk about the experimental procedure *via* which we try to examine arbitrary decision making in autism.

## Dual-system model and arbitrary decision making

In our everyday life, when confronted with dilemmas that can be analysed with rational criteria, either one or both thinking systems can be recruited to reach a decision (*Kahneman, 2003*). What happens however, when arbitrary decisions need to be made? In situations where arbitrary decisions need to be made no clear reason exists as to why one choice should be preferred over the other one, since from all available choices none is more advantageous than the others. This scenario might seem extreme but almost every day we are faced with such choices. Which apple would you pick from the pile in the store, when all apples placed in front of you are of the same quality? When exactly will you choose to schedule your first appointment, during working hours, when your calendar is free, at 10:15 am or 10:20 am? Which pencil would you peak from a box of identical pencils? These and many other similar decisions raise the question of how a "thinking" system that is based on reason, can actually make such arbitrary choices?

Depending only on system 1, an arbitrary decision can be made automatically, without active consideration, and it could be 'biased' or not. By 'biased' we mean any decision that would not be considered random by the system. System 1, for example, may be guided from our habits to make such a decision. In that case and in the aforementioned example we would choose the apple that is on the left side of the pile, because we always do so but may not consciously remember. When not 'biased', our arbitrary decisions could be the result of a random function that could be expressed *per se* or influence system 1 (*Hogarth, 2005*).

When the slower system 2 is involved in arbitrary decisions, this can happen through aroused emotions, memories and attention to environmental nudges that are provoked by system 1 (*e.g.*, recognize the outline of a beloved person on the surface of the apple), by substituting the question we are faced with, with a rational one that could be answered easily (*e.g.*, which apple is more red), by consciously following or deviating from a set of rules (*e.g.*, always choose apples from the bottom row or decide to actively avoid this pattern), or by depending on our environment either passively (*e.g.*, pick the apples that will remain untouched by other customers) or actively (*e.g.*, coin flipping or asking someone else to make a decision for us) (*Hogarth, 2001*; *Moulton et al., 2007*). Our efficiency and ability to make such decisions, can impact our everyday functioning. If we actively need to consider all our choices, that would require the involvement of system 2 and this could impact our performance especially in relation to the time and effort we spend in decision making (*Croskerry & Nimmo, 2011*; *Moulton et al., 2007*). Moreover, if the responsibility of making a decision has to be transferred to someone else, then that will result in a dependent

behaviour. Alternatively, when certain rules need to be strictly followed that can lead to a stereotypical behaviour.

## Randomness in arbitrary decision making

Randomness as a concept is by itself a complicated one, and the way humans perceive randomness might differ from the actual physical notion of the concept (*Bar-Hillel & Wagenaar, 1991*). Many attempts to define randomness have been made. Each one focuses on different concepts that events which could be considered random should entail, including equal probability, sequential independence and unpredictability (*Nickerson, 2002*). As the most widely accepted characteristic of a random sequence of events would be that of equal probability, for the purpose of this paper, we define a sequence of infinite events as random when all possible events have equal probability to occur at each time point of this sequence (*Borowski, n.d*).

Several theories have been stated and experiments conducted in order to examine how humans perceive randomness. Many of these theories and experiments support that humans are very poor in recognizing sequences that are the result of random processes (for review of earlier experiments *Tune, 1964*). When the production of random number sequences is concerned, literature findings are conflicting as some studies show that they produce number sequences that very well resemble those of a random process (*Persaud, 2005*) while other studies indicate that humans have a relatively poor performance in this task (*e.g.*, *Bains, 2008*; *Figurska, Stańczyk & Kulesza, 2008*). Some of these findings could be due to the fact that our cognitive capacity is limited (*Bains, 2008*; *Persaud, 2005*). Moreover, when no instructions are provided to participants as to what is considered random or no indication is provided that a random approach would be needed to execute a task, participants can produce random sequences approaching them as arbitrary decisions and thus successfully engage in tasks where a random approach is the best strategy (*Rapoport & Budescu, 1992*; *Sharifian, 2016*).

Randomness exists in nature and can behaviourally be expressed as an ability to avoid predictability, thus linked to an evolutionary advantage (*Riotte-Lambert & Matthiopoulos, 2020*) . Experiments in animals have shown that their brain can turn on a 'random mode' when rules and settings change and there is no prior knowledge to guide decisions (*Tervo et al., 2014*). In the same context, when infants are placed in a novel environment, for example with toys they have never seen before, the decision around which one they would pick, is arbitrary. To reach that decision, they have to obviously recruit a randomness mechanism. The choice they make, under an evolutionary perspective, if proven a safe one, would shape their future preferences (*Silver et al., 2020*).

Randomness in decision making is a rather difficult concept to examine experimentally. We can identify three ways randomness can be expressed in arbitrary decision making. True randomness when uncontrolled and unpredictable phenomena are utilized (*e.g.*, coin flipping, rolling dice), imitated randomness when decisions are made based on a set of rules (*e.g.*, making a conscious unprecedented choice to avoid following a pattern) and innate randomness through a possible innate mechanism of randomness production. Finally, a combination of the last two ways of randomness expression could serve arbitrary decision

making, especially when a set of rules is not fully descriptive. For example, a decision to produce a number that cannot be a repetition of the last two numbers produced, can comply both with imitated and innate randomness. The first two ways, rely heavily on system 2 while the last mechanism would be part of system 1. The concept of innate randomness is one proposed by the authors of this paper, based on previous research findings and observations (for example, *Wong, Merholz & Maoz, 2021*; *Tervo et al., 2014*; *Schulz et al., 2012*; *Laskaris, Zafeiriou & Garefa, 2009*; *Tune, 1964*). The characteristics of an innate randomness mechanism remain largely unknown, although some observations have been made in animal and human studies (*Tervo et al., 2014*). In this study, we try to examine the possibility of innate randomness existence by trying to "force" its expression by favoring system 1.

## Autism and the dual-system thinking model

Autism is one of the most common neurodevelopmental disorders (*CDC, 2016*). Its symptomatology includes persistent impairments in reciprocal communication and social interactions as well as restricted, repetitive patterns of behaviours, interests or activities (*APA, 2013*). Different autism phenotypes are characterized by a diversity of traits (*Geschwind, 2009*).

As far as decision making is concerned, autistic individuals are faced with difficulties, and are generally slower in taking decision and show less intuitive and more deliberate reasoning (*Brosnan, Lewton & Ashwin, 2016*; *Farmer, Baron-Cohen & Skylark, 2017*; *Luke et al., 2012*; *Vella et al., 2018*). To account for those characteristics and based on experimental knowledge, *De Martino et al. (2008)* proposed a model according to which autism would more heavily rely on system 2 when decision making is concerned.

Although helpful to interpret certain autistic behaviours during decision making, the De Martino model, does not offer any insight concerning the role of system 1. It is possible that autistic individuals heavily rely on system 2 as a counterbalancing strategy to a compromised system 1. Alternatively, system 1 could be intact but system 2 is overactive and dominant. Additionally, autistic individuals experience difficulties in using emotional or attentional cues in order to make decisions (*Gaigg, 2012*), indicating that the way by which system 1 could influence system 2 is even further limited and this could lead to stereotypical behaviour (*Cunningham & Schreibman, 2008*; *Farmer, Baron-Cohen & Skylark, 2017*) and dependency and would point towards autism being characterized by difficulties in abstract decision making. Reports of autistic individuals and their caregivers highlight the difficulties of these individuals to decide, especially when new choices need to be made or when decisions need to be taken on the spot (*Frith, 2001*; *Vella et al., 2018*). A recent study has indeed shown that autistic individuals that have high functioning ASD experience difficulties in routine everyday decisions, including what clothes to wear and when to shower but these difficulties are not present when crucial life decisions are involved (*Levin et al., 2015*). This difficulty could be a limiting factor to their everyday functionality which is of paramount importance and a major factor to consider in the diagnosis of autism.

## The Random Number Generation (RNG) task

The RNG task is commonly used to examine the concept of randomness in humans (*Matsukawa et al., 2006*; *Rosenberg et al., 1990*; *Shinba et al., 2000*; *Spatt & Goldenberg, 1993*; *Williams et al., 2002*). During the task, participants are asked to produce a sequence of numbers that adheres to predefined rules that would make a sequence random, such as avoid patterns, repetitions of the same digits or use all digits available before recycling them. This could lead to responses that would imitate randomness. All the different ways that the RNG task has been employed, are based on the definition of what constitutes a random number sequence, that is either provided to participants or is based on how they perceive randomness themselves.

In that sense, the RNG and similar tasks, become heavily dependent on cognitive and attentional processes (*Towse, 1998*; *Williams et al., 2002*). Each choice of number is not an entirely arbitrary choice but a well thought decision which would depend on processes that are employed by system 2. In these types of experiments, participants' performance improves with training and feedback, however, these experiments deviate from a concept of innate randomness and treat randomness as a skill that could be improved with practice (*Biesaga, Talaga & Nowak, 2021*; *Gauvrit et al., 2017*). Indeed, a recent experiment by *Biesaga, Talaga & Nowak (2021)* has shown that when cognitive demands and contextual task conditions are optimal, participants can produce sequences that could be considered more random, highlighting the importance of providing task instructions to participants to achieve that effect. The cognitive demands of such tasks become apparent however, as participants tire with time and the randomness of the sequences they produce decreases. The significant contribution of cognitive processes to the typical RNG task, also becomes apparent when the task is performed by individuals who experience cognitive difficulties. Patients with Alzheimer's disease, schizophrenia as well as autistic individuals (low functioning) tend to produce shorter number sequences with more digit cycling and repetitions than the comparison groups and these differences have been attributed to attentional deficits, difficulties in working memory and executive functions (*Brugger et al., 1996*; *Proios, Asaridou & Brugger, 2008*; *Williams et al., 2002*).

## Study aims

The primary aim of this exploratory study is by silencing system 2, to show the contribution of system 1 and innate randomness in arbitrary decision making in autism. In order to achieve that, we modified the RNG task to limit the contribution of cognitive processes, such as working memory and consequently the contribution of system 2 to the task. That would allow us to examine the role of system 1 in arbitrary decision making in autism. The RNG task has not, to our knowledge, been administered to autistic participants with high functioning ASD before. However, given that autistic individuals rely heavily on system 2 to make decisions (*De Martino et al., 2008*; *Levin et al., 2015*) and considering that in our task the contribution of system 2 is reduced, we hypothesize that the autism and the comparison group would differ in both the quality of the randomness features and the quantity of the numbers produced.

**Table 1  Demographics.** Summary of the age, gender balance and IQ scores of the ASD and the comparison groups.

|  | ASD | Control | p-value[*] |
|---|---|---|---|
| **Age (years) mean** (std) | 27 (6.8) | 26 (5.7) | 0.368 |
| **Gender n** (%) |  |  | 0.842 |
| Male | 30 (84) | 34 (85) |  |
| Female | 6 (16) | 6 (15) |  |
| **IQ mean** (std) | 107 (11.6) | 111 (10.8) | 0.077 |

**Notes.**

*Mann–Whitney U was used when non-normality could not be excluded, else ANOVA. For Gender% comparison Chi-square was used.

# METHODS

## Participants

This study recruited autistic adults (36 adults, 30 male) and an age and IQ-matched comparison group (40 adults, 34 male). For details see Table 1.

The autistic participants were recruited from a larger cohort of volunteers (*Pehlivanidis et al., 2020*) who participated in a research project on the *de novo* diagnosis of adults with neurodevelopmental disorders. ASD diagnosis was based on DSM-5 criteria and everyone in the autism group had fluent phrase speech and more than 12 years of education. Exclusion criteria included the presence of acute psychopathology, systematic psychopharmacological treatment up to 30 days prior to taking part in the study, current substance abuse disorder, IQ<70 assessed with Wechsler Adult Intelligence Scale (WAIS-IV), any known genetic condition (*Wechsler, 2012*).

The comparison group was recruited *via* advertisement and word of mouth and included neurotypical individuals with IQ>70 (WAIS-IV) and more than 12 years of education. Exclusion criteria included the presence of acute psychopathology.

Written informed consent was obtained from all participants and the study was approved by the Ethics Committee of the Department of Psychiatry, National and Kapodistrian University of Athens (IRB 12/7/2018 #517). All experiments were performed in accordance to relevant guidelines and regulations.

## Modified random number generation task

Participants were instructed to pronounce a sequence of numbers, as comes to mind, with each number ranging from 1 to 10. The task duration was 1 min. Participants were advised but not obliged to produce numbers at a steady pace of approximately 1 number per second. No example sequence was given to participants to avoid unwanted influences from the experimenter.

To ensure minimal distractions and keep the experimental conditions as similar as possible between participants, individuals were placed in a dark room. The only source of light was a computer screen which was displaying a progress bar used to count down the remaining task time and helping participants to keep the pace. Throughout the task, participants were listening to a track consisting of nature sounds (forest, rain sounds etc.) at a loud volume through a headset. Nature sounds were chosen as they already are

familiar to individuals. The sound volume was loud enough to block any surrounding noise as well as the sound of their own voice and this was confirmed before the start of the task. The purpose of this modification was to limit the contribution of auditory working memory—a process mainly employed by system 2, in other words limit the ability of individuals to choose a number based on the sequence of numbers that have already been used. The sequence was recorded automatically. The experimenter was present in the room to ensure proper execution of the task but was out of the participants' field of view and did not interact with them in any way. Each participant produced a single random number sequence and at the end of each session the number sequences were transcribed by the investigator and the participants' recording were deleted.

## Data analysis

The task data were processed using in house code in R and Python programming languages. Two groups of variables were calculated using the RandSeq (*Oomens et al., 2021*) package in R. The first group of variables included standard measures of randomness: Redundancy (R), Random Number Generation (RNG), RNG2, Null Score Quotient (NSQ), Adjacency (see Table S1). The second group included measures that describe the characteristics of the sequence itself: response frequencies (RF), first order difference (FOD) (*Towse, 1998*) as well as the correlation frequency index (Cf) (*Barbasz et al., 2008*) (Table 2). Given that in our task sequence lengths were different between participants we computed the relative equivalent of the second group of variables (Table 2). These are standard variables that have been widely used in the literature to examine human randomness as expressed through the RNG task. In order to observe how randomness measures might change over time, an overlapping moving window of 20 secs in length (33% of total task time) was also applied to the sequences and the above variables were calculated for each step. In order to focus more on any "pattern" characteristics that the task sequences might have we also performed a recurrence quantification analysis (RQA) in R using casnet package (*Hasselman, 2022*). All the variables that were calculated are defined in Table 1 and for the values of each variable for both groups are summarised in Table 2.

A classification approach was used to examine whether we can distinguish the two groups based on the characteristics of the sequences that each group produced. The sklearn Python package for machine learning was used and a decision tree method was followed. The GridSearchCV was consulted for the hyperparameters selection. The model was tested for all possible combinations of features, up to three features and leave-one-out cross validation was applied. For the replication of the results the 'random_state' variable in both splitting the data and the decision tree functions, was set to 1.

## RESULTS

The lengths of the number sequences produced during the modified RNG task, did not differ between the autism group and the comparison group ($p = 0.27$, comparison group: mean = 62, median =58, IQR = 49−72, autism group: mean = 59, median = 54, IQR = 40−74). Based on the characteristics of the sequence, both the autism and the comparison group showed similar profiles in the use of pairs of consecutive numbers and the frequency

**Table 2  Variables description.** For the analysis of the RNG task three groups of variables were calculated. The variables that belong to each group and their definitions are offered here.

| | Variable Name | Variable Definition |
|---|---|---|
| | Redundancy (R) | The measure of inequality in response which determines the extent of deviation from ideal information generation. |
| | Random Number Generation (RNG) | The measure of inequality of response usage between observed and mathematical digrams for adjacent responses. |
| | Random Number Generation 2 (RNG2) | The measure of inequality of response usage between observed and mathematical digrams for interleaved responses. |
| Group 1 Measures of Randomness | Null Score Quotient (NSQ) | The measure of inequality of response usage between digrams that do not appear in the sequence for adjacent responses. |
| | Adjacency | The measure of frequency of digrams of adjacent responses |
| | Response Frequency (RF) | The measure of occurrence of each response per sequence |
| | Relative Response Frequency (r_RF) | The measure of occurrence of each response per sequence length |
| | First Order Difference (FOD) | The measure of occurrence of the arithmetic difference between each response and its preceding value |
| Group 2 Characteristics of the sequence | Relative First Order Difference (r_FOD) | The measure of occurrence of the arithmetic difference between each response and its preceding value per sequence length |
| | Correlation function index (Cf) | The measure of occurrence of repetition of the same digit for all possible distances |
| | Relative correlation function index (r_Cf) | The measure of occurrence of repetition of the same digit for all possible distances, divided by the maximum possible Cf value, for the sequence length |
| | Recurrence Rate | The percentage of recurrent responses in the sequence |
| | Determinism | The measure of quantification of repetitive patterns |
| | Laminarity | The measure that represents the proportion of digits that are repeated monotonously, without any time or digit interval |
| Recurrence Quantification Analysis | Shannon information entropy | The measure of the complexity of the deterministic structures of the sequence |
| | Trapping time | The measure of the average length of repetitive patterns of digits that are repeated monotonously |
| | Mean | The measure of average length of repetitive patterns |

of digits (Fig. 1). When the randomness measures were calculated and compared between the two groups, few significant differences were identified between the two groups for some of the variables in both groups. For a detailed summary of these results see Table 3.

In the next step, the 20 secs moving window was applied to the data and the randomness values were plotted for each step. We observe that during the task's time course, the differences of the randomness values between the two groups were very small, however, for three of these measures this difference constantly remained in favor of one group (Fig. 2). Specifically, the NSQ and redundancy indices were borderline higher for the autism group while the adjacency combined values were higher for the comparison group.
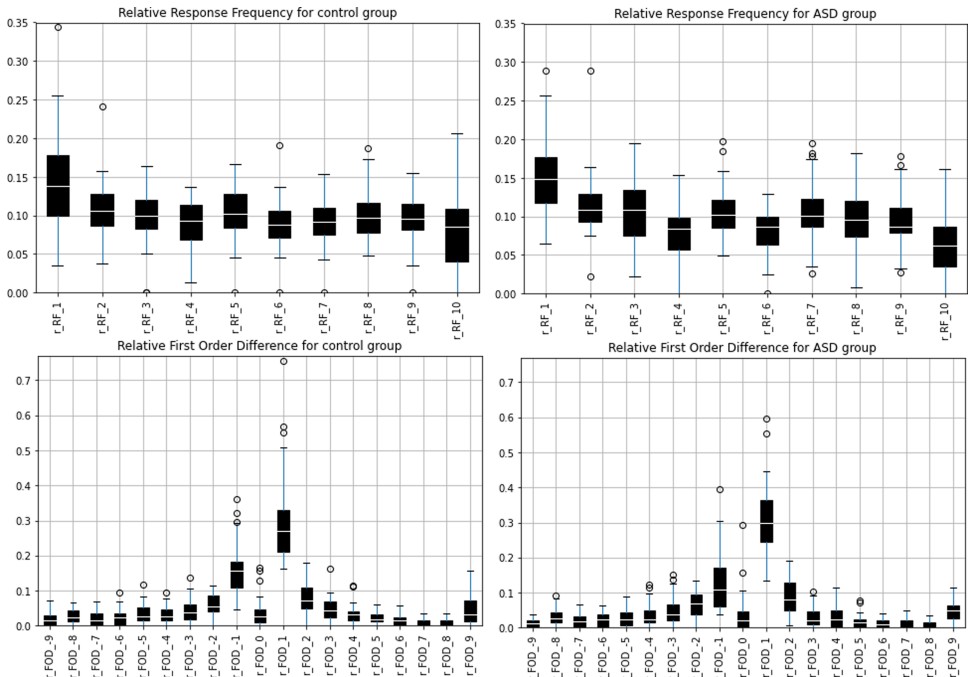

**Figure 1** **The relative FOD and relative RF variables by group.** The relative FOD and relative RF variables as were calculated for the autism group and the comparison group. Both random number sequences show similar patterns in the use of consecutive pairs (r_FOD) and the frequency of digits (r_RF).

No significant differences were identified between autism and the comparison group when the RQA approach was applied to the data.

Finally, for the decision tree method, when the descriptive measures of randomness were used as features for classification, there were more than one combinations of up to three features that could discriminate the groups with relatively high mean accuracy rate (>75%). In Fig. 3, we present the decision tree for a two features model, FOD_8 and FOD_-8 (hyperparameters: max_depth =5, max_features =1, min_samples_split =7, min_samples_leaf =1) where the mean classification accuracy reached 82%.

## DISCUSSION

The primary aim of this exploratory study is by silencing system 2, to show the contribution of system 1 and innate randomness in arbitrary decision making in autism. For that purpose, we recruited a sample of autistic individuals with high functioning ASD and a comparison group, who performed a modified version of the RNG task. Our version differed from the typical RNG in that the task duration was kept the same for all participants who were placed in a dark room and were listening to nature sounds at a very high volume while producing the random number sequence. No definition of randomness and rules for the production of number sequences were given. Matching our experiment's characteristics with Wanegaar's three features for random processes (*Wagenaar, 1991*), we provided participants with a "fixed set of alternatives" (digits from 1 to 10) and tried to make the

**Table 3  Summary of the variables per group.** Summary of the variables of the sequences for the ASD group and the comparison group. For the second group of variables, their relative equivalent was also calculated since our sequences where not equal in length. The relative values are also included in the table.

| Group | Description | Name | ASD mean (std) | Ctrl mean (std) | *p* value[*] |
|---|---|---|---|---|---|
| Measures of Randomness | The measure of inequality of response usage between observed and mathematical digrams for adjacent responses | RNG | 42.095 (16.82) | 45.16 (18.41) | 0.414 |
| | The measure of inequality of response usage between observed and mathematical digrams for interleaved responses. | RNG2 | 35.739 (14.62) | 35.398 (16.32) | 0.839 |
| | The measure of inequality of response usage between digrams that do not appear in the sequence for adjacent responses. | NSQ | 66.778 (8.45) | 64.878 (7.71) | 0.093 |
| | Measure of inequality in response, which determines the extent of deviation from ideal information generation. | Redundancy | 4.405 (3.8) | 4.022 (5.01) | 0.31 |
| | The measure of frequency of digrams of adjacent responses | Adjacency | 34.461 (16.23) | 36.37 (19.23) | 0.892 |
| | | TPI | 80.486 (21.48) | 81.08 (21.62) | 0.65 |
| | | RF_1 | 9.277 (6.38) | 8.525 (3.48) | 0.714 |
| | | RF_2 | 7.194 (5.7) | 6.375 (2.68) | 0.937 |
| | | RF_3 | 6.75 (4.53) | 6.35 (3.3) | 0.862 |
| | | RF_4 | 4.75 (2.58) | 5.725 (2.88) | 0.189 |
| | The measure of occurrence of each response per sequence | RF_5 | 6.222 (3.08) | 6.525 (3.2) | 0.55 |
| | | RF_6 | 4.861 (2.36) | 5.7 (3.19) | 0.343 |
| | | RF_7 | 6 (2.72) | 5.925 (2.86) | 0.689 |
| | | RF_8 | 5.527 (3.12) | 6.325 (3.22) | 0.306 |
| | | RF_9 | 5.305 (2.16) | 5.925 (2.34) | 0.16 |
| | | RF_10 | 3.472 (2.26) | 4.725 (3.02) | 0.032 |
| | | r_RF_1 | 0.151 (0.04) | 0.144 (0.06) | 0.589 |
| | | r_RF_2 | 0.114 (0.04) | 0.104 (0.03) | 0.432 |
| | | r_RF_3 | 0.107 (0.04) | 0.099 (0.03) | 0.472 |
| | | r_RF_4 | 0.08 (0.03) | 0.09 (0.028) | 0.145 |
| | The measure of occurrence of each response per sequence length | r_RF_5 | 0.106 (0.03) | 0.103 (0.032) | 0.741 |
| | | r_RF_6 | 0.082 (0.02) | 0.087 (0.03) | 0.425 |
| | | r_RF_7 | 0.103 (0.03) | 0.093 (0.02) | 0.173 |
| | | r_RF_8 | 0.095 (0.03) | 0.099 (0.03) | 0.619 |
| | | r_RF_9 | 0.095 (0.03) | 0.096 (0.03) | 0.881 |
| | | r_RF_10 | 0.062 (0.03) | 0.079 (0.04) | 0.094 |
| | | FOD_-9 | 0.972 (1.1) | 1.375 (1.68) | 0.418 |
| | | FOD_-8 | 2.027 (1368) | 1.75 (1.37) | 0.681 |
| | | FOD_-7 | 1.361 (1.43) | 1.475 (1.55) | 0.888 |
| | | FOD_-6 | 1.5 (1.25) | 1.65 (1.45) | 0.776 |
| | | FOD_-5 | 1.833 (1.87) | 2.125 (1.62) | 0.269 |
| | | FOD_-4 | 2.222 (1.88) | 2.025 (1.64) | 0.758 |
| | | FOD_-3 | 2.916 (2.07) | 2.625 (1.7) | 0.755 |

**Table 3** (*continued*)

| Group | Description | Name | ASD mean (std) | Ctrl mean (std) | $p$ value[*] |
|---|---|---|---|---|---|
| Characteristics of the sequence | The measure of occurrence of the arithmetic difference between each response and its preceding value | FOD_-2 | 4.333 (2.54) | 4.175 (2.36) | 0.779 |
| | | FOD_-1 | 9.805 (10.13) | 11.525 (6.27) | 0.015 |
| | | FOD_0 | 3.361 (6.97) | 2.6 (3.05) | 0.395 |
| | | FOD_1 | 21.722 (10.69) | 22.2 (16.28) | 0.646 |
| | | FOD_2 | 5.944 (3.98) | 5.65 (3.54) | 0.916 |
| | | FOD_3 | 1.944 (1.83) | 3.275 (2.45) | 0.008 |
| | | FOD_4 | 2.027 (2) | 2.225 (1.68) | 0.424 |
| | | FOD_5 | 1.166 (1.18) | 1.475 (1.06) | 0.106 |
| | | FOD_6 | 0.861 (0.96) | 1 (1.01) | 0.551 |
| | | FOD_7 | 0.75 (0.9) | 0.6 (0.74) | 0.577 |
| | | FOD_8 | 0.416 (0.64) | 0.475 (0.71) | 0.798 |
| | | FOD_9 | 3.194 (2.36) | 2.875 (2.62) | 0.471 |
| | | r_FOD_-9 | 0.013 (0.01) | 0.018 (0.02) | 0.45 |
| | | r_FOD_-8 | 0.03 (0.02) | 0.026 (0.02) | 0.392 |
| | | r_FOD_-7 | 0.02 (0.01) | 0.02 (0.02) | 0.849 |
| | | r_FOD_-6 | 0.023 (0.01) | 0.024 (0.02) | 0.945 |
| | | r_FOD_-5 | 0.028 (0.02) | 0.032 (0.02) | 0.537 |
| | | r_FOD_-4 | 0.036 (0.03) | 0.03 (0.02) | 0.754 |
| | | r_FOD_-3 | 0.046 (0.03) | 0.04 (0.02) | 0.522 |
| | | r_FOD_-2 | 0.067 (0.03) | 0.06 (0.03) | 0.333 |
| | The measure of occurrence of the arithmetic difference between each response and its preceding value per sequence length | r_FOD_-1 | 0.128 (0.08) | 0.159 (0.07) | 0.025 |
| | | r_FOD_0 | 0.037 (0.05) | 0.035 (0.03) | 0.67 |
| | | r_FOD_1 | 0.317 (0.09) | 0.295 (0.12) | 0.083 |
| | | r_FOD_2 | 0.088 (0.05) | 0.079 (0.04) | 0.45 |
| | | r_FOD_3 | 0.028 (0.02) | 0.046 (0.03) | 0.014 |
| | | r_FOD_4 | 0.033 (0.03) | 0.031 (0.02) | 0.979 |
| | | r_FOD_5 | 0.018 (0.01) | 0.021 (0.01) | 0.257 |
| | | r_FOD_6 | 0.012 (0.01) | 0.015 (0.01) | 0.607 |
| | | r_FOD_7 | 0.012 (0.01) | 0.009 (0.01) | 0.544 |
| | | r_FOD_8 | 0.007 (0.01) | 0.007 (0.01) | 0.916 |
| | | r_FOD_9 | 0.046 (0.02) | 0.044 (0.04) | 0.5 |
| | The measure of occurrence of repetition of the same digit for all possible distances | Cf | 3.507 (1.95) | 3.494 (1.09) | 0.213 |
| | The measure of occurrence of repetition of the same digit for all possible distances, divided by the maximum possible Cf value, for the sequence length | r_Cf | 0.099 (0.01) | 0.098 (0.01) | 0.946 |
| | The percentage of recurrent responses in the sequence | RR | 0.101 (0.01) | 0.1 (0.01) | 0.847 |
| | The measure of quantification of repetitive patterns | DET | 0.317 (0.09) | 0.337 (0.12) | 0.669 |
| | The measure that represents the proportion of digits that are repeated monotonously, without any time or digit interval | LAM_hl | 0.076 (0.12) | 0.071 (0.08) | 0.56 |

**Table 3** (*continued*)

| Group | Description | Name | ASD mean (std) | Ctrl mean (std) | *p* value[*] |
|---|---|---|---|---|---|
| Recurrence Quantification Analysis | The measure of average length of repetitive patterns | MEAN_dl | 2.628 (1.4) | 2.753 (2.47) | 0.818 |
| | The measure of the complexity of the deterministic structures of the sequence | ENTrel_dl | 0.147 (0.08) | 0.149 (0.09) | 0.75 |
| | The measure of the average length of repetitive patterns of digits that are repeated monotonously | TT_hl | 1.344 (1.03) | 1.479 (0.93) | 0.99 |

**Notes.**
[*]Mann–Whitney U was used when non-normality could not be excluded, else ANOVA.

digit selection a "memoriless procedure", by mitigating working memory through the very loud noise manipulation. Finally, by removing the task instructions and feedback, we tried to make each digit choice an abstract decision thus removing "preference for any of the possible outcomes".

The task data were analyzed for both groups and we compared the descriptive characteristics of the sequences, the standard measures of randomness that are calculated when the RNG task is analyzed and the output indices of the RQA analysis. We did not identify any differences in the above measures between the two groups with the exception of certain isolated descriptive measures of the number sequence, namely the use of consecutive numbers with a difference of −1 or +3, FOD_-1 and FOD_3, that were preferred by the autism group.

Our findings are not in line with our initial hypothesis as we were expecting the autism group to show reduced production and consequently, sequences that are shorter in length and less random compared to the comparison group (*Rinehart et al., 2006*; *Williams et al., 2002*). A more repetitive and less original number sequence would have been in line not only with RNG literature but also research indicating that autistic children produce less complex and less original color and tone patterns, compared to their non-autistic peers. Although these color and tone experiments were not conducted with the aim to investigate the concept of randomness in autism, they indeed indicate that autistic individuals tend to adhere more strictly to rules and patterns when faced with tasks that resemble the RNG.

Our findings are also in contrast to a study by *Williams et al. (2002)* which showed that the autism group produced number sequences with more digit cycling and patterns in relation to the comparison group. This study, however, recruited a group of autistic participants with low functioning ASD who were asked to perform a typical RNG task during which they were instructed to avoid patterns and repetitions of numbers. The differences in the number sequences that were produced, were attributed to the cognitive characteristics of the autism group, and specifically to impaired response inhibition that contributes to participants repeating digits. Such a finding is in line with the RNG recruiting cognitive processes that primarily are employed by system 2 (*Brugger et al., 1996*).

In our RNG task and in order to look at how the randomness measures fluctuate over time we applied an overlapping moving window and the randomness measures were calculated for every window step. We did not identify any significant fluctuations of randomness measures over time. Although NSQ, redundancy and adjacency indices were
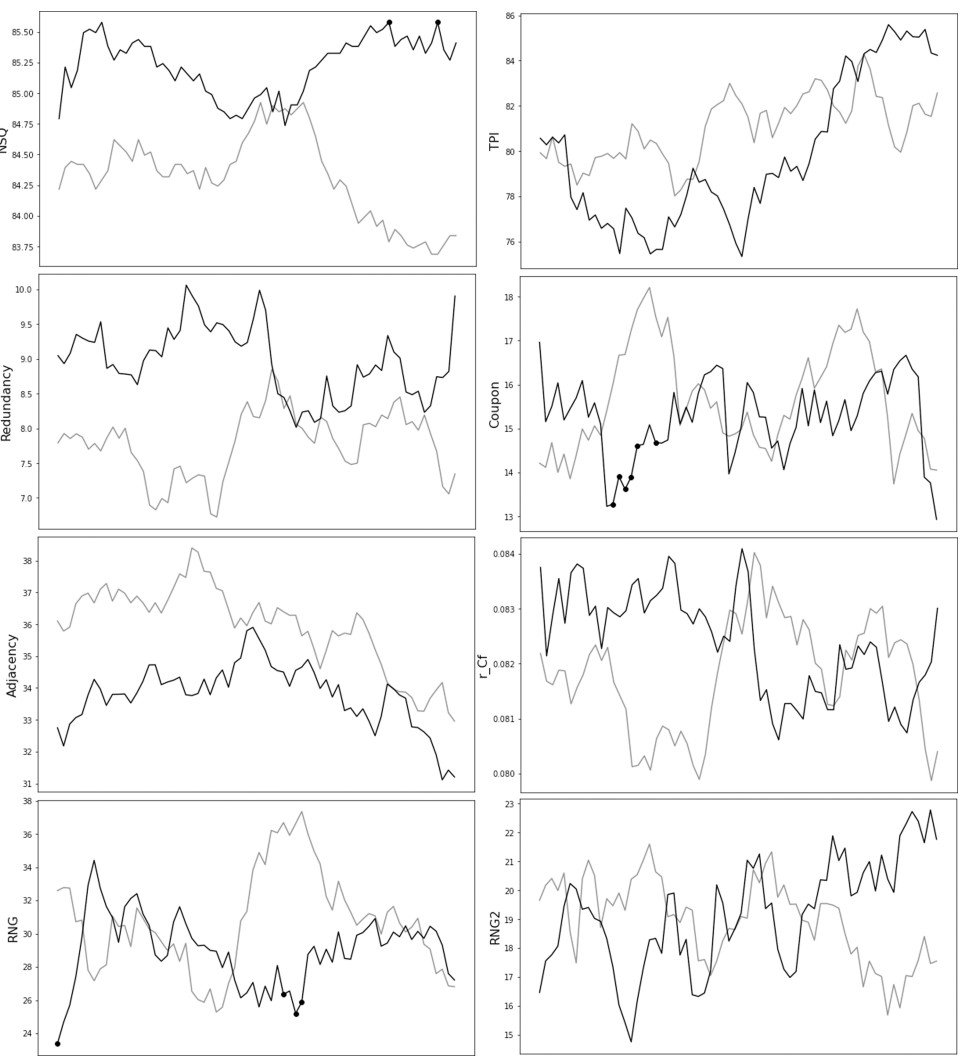

**Figure 2 Randomness measures' 20 secs moving window.** Randomness measures that differed between the autistic group and the comparison group, for a 20secs moving window. A 20 s moving window was applied to the data to examine how randomness measures develop over time, in each group. The values for the autistic group are represented with the dark grey line, while the control group values are depicted using a light grey line.

almost always in favor of one group, the difference of those measures between the two groups was not big enough to yield any significant conclusions.

When machine learning was used and the decision tree approach was applied to the data we were able to discriminate between the autism and the comparison group with a classification accuracy that reached 82%. We could offer two different viewpoints for this finding. It is possible that there are differentiating patterns in the number sequences of the two groups that could not be picked up when the standard randomness indices were compared one by one, but were reflected when randomness measures were viewed in combination. Alternatively, both groups could be equally random but random in a

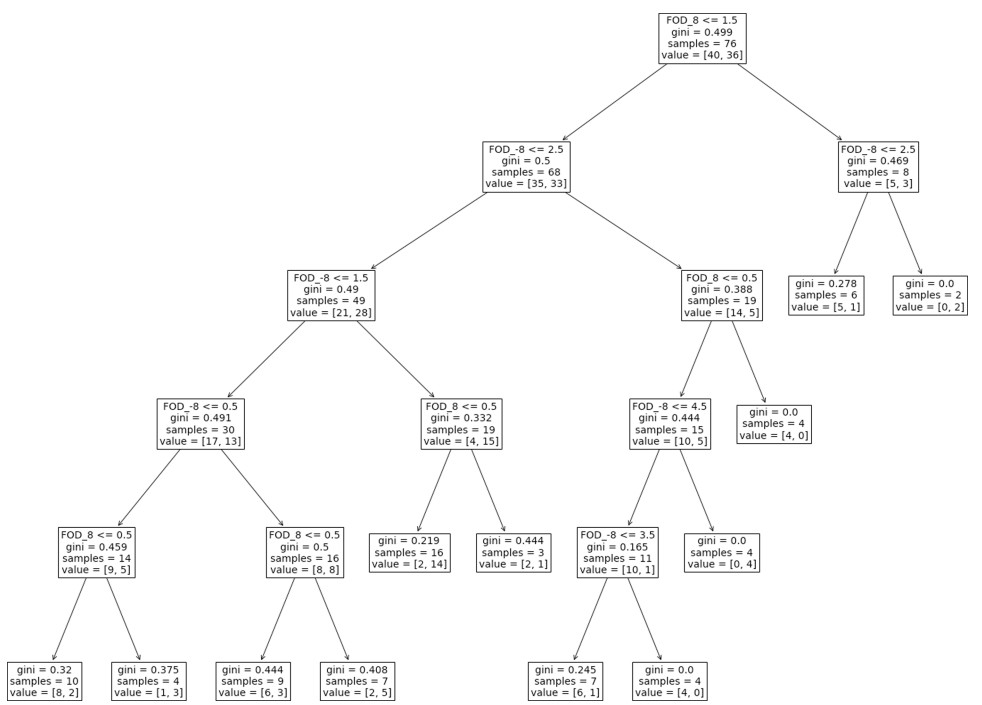

**Figure 3 Machine learning decision tree.** The decision tree for the two variables, FOD_8 and FOD_-8, that returned a classification accuracy of 82% (hyperparameters: max_depth =5, max_features =1, min_samples_split =7, min_samples_leaf =1).

different way, for example number sequences 1 2 3 2 3 and 4 5 6 5 6 are different but equally random and they even contain the same amount of repetitions.

Recent data suggest that individuals' responses during random sequence generation could be considered a fingerprint—a unique biomarker for the individuals' cognitive ability—and it could also vary between psychopathological populations (*Laskaris, Zafeiriou & Garefa, 2009*; *Schulz et al., 2012*; *Wong, Merholz & Maoz, 2021*). Specifically, several studies have used a classification approach and they have shown that the mechanism by which individuals generate random sequences is highly unique and consistent enough to allow a model to predict the next item in the sequence for each individual subject (*Schulz et al., 2012*; *Schulz et al., 2020*). Additionally, *Jokar & Mikaili (2012)* examine the randomness of the temporal characteristics of a sequence which appear unique enough to discriminate between subjects.

Our study indicates the possibility that the individual patterns during random sequence production are consistent enough between groups to allow for an accurate discrimination between a clinical population and the comparison group, using machine learning algorithms. As our study aimed to explore a novel concept, that of innate randomness, further research and replication of our findings is paramount. This would allow us to better understand how innate randomness is expressed during arbitrary decision making and how its mechanisms might present with altered function in autism as well as other clinical populations. Our findings, if confirmed by future studies and expanded to other

clinical populations, would indicate that tasks such as the RNG which are likely to recruit randomness mechanisms, could be useful diagnostic tools and could help expand our understanding of the unique characteristics of several psychopathological populations.

The study has several limitations. First, concerning the characteristics of our sample, no information was gathered around our ASD group on whether they had access to some form of psychotherapy. Despite adjusting our experimental conditions, we cannot exclude the contribution of system 2 during the RNG task. Specifically, this would be in line with experiments. Moreover, the RNG task has not been used before in autistic individuals with high functioning ASD and thus we cannot test the contribution of our manipulations in our group's performance. In our version of the RNG task, as well as the standard task version, participants are asked to produce random sequences using numbers from 1 to 10. The use of sequential numbers, that can be arranged based on their value and with which we are familiar since very early in life, would favor biases for specific numbers and number sequences. Indeed, in our groups we do see a consistent preference for specific digits, that could be due to cultural or environmental influences. This may limit the expressed randomness and could partly account for the absence of significant differences in our groups. To overcome some of these issues, a bigger sample and/or the use of an alternative task could be of help. Ideally, a task that would minimize such cultural and environmental influences might be more appropriate to examine the randomness characteristics of different groups and individuals. Our classification results need further validation since we cannot attest to the specificity of our findings for autism. Given our small sample and the hyperparameters (min_samples_leaf $=1$) that we used for classification, there is always the danger that our high classification accuracies were due to overfitting. A bigger sample would allow us to further explore the possibility of discrimination between different groups, based on their produced sequences.

## ACKNOWLEDGEMENTS

We would like to thank our mothers, Venetia Manta and Maria Athanasarou, for their invaluable support for bringing this work into completion.

### Funding

The authors received no funding for this work.

### Competing Interests

The authors declare there are no competing interests.

### Author Contributions

- Vasileios Mantas conceived and designed the experiments, performed the experiments, analyzed the data, prepared figures and/or tables, authored or reviewed drafts of the article, and approved the final draft.

- Vasileia Kotoula analyzed the data, prepared figures and/or tables, authored or reviewed drafts of the article, and approved the final draft.
- Artemios Pehlivanidis performed the experiments, authored or reviewed drafts of the article, and approved the final draft.

## Human Ethics

The following information was supplied relating to ethical approvals (*i.e.*, approving body and any reference numbers):

The Ethics Committee of the Department of Psychiatry, National and Kapodistrian University of Athens approved the study (12/7/2018 #517).

## Data Availability

The raw data are available in the Supplemental File.

## Supplemental Information

Supplemental information for this article can be found online at http://dx.doi.org/10.7717/peerj.15751#supplemental-information.

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
