# Peer review of "Exploring randomness in autism"

_PeerJ, doi:10.7717/peerj.15751_

## Round 0.1 · original submission · Major Revisions

I have now received the reviewers' comments on your manuscript. They have suggested some revisions to your manuscript. Therefore, I invite you to respond to the reviewers' comments and revise your manuscript.

Reviewer 1 ·

Basic reporting

In this paper the authors investigate how, a novel concept that of innate randomness, could be expressed in ASD, in situations where arbitrary decision making is needed. To do that, the authors employ the RNG task but modify it, in order to limit the contribution of system 2 and allow system 1 and innate randomness to emerge.

The concept of innate randomness is presented thoroughly in the introduction and the authors review the literature around the RNG task as well as decision making in ASD rather nicely. However, the paper and especially the introduction are specifically hard to read through, especially at certain points. The modification that the authors apply to the task is also an interesting one while the discrimination of the ASD group from the control group is a unique finding that highlights the importance of multivariate approaches when such data are considered. As the discussion underlines, however, the paper has several weakness and replication is needed before firm conclusions could be drawn around the existence of innate randomness and arbitrary decision making in ASD.

1.The language used to describe autistics does not always adhere to the preferences of the autistic community. Could you please revise the paper so that it is in accordance to the terminology that is currently adopted by the scientific community to describe ASD?
For example, in line 153 the authors use the term “ ASD patients” which is not acceptable as autism is not considered an illness and it is preferable to refer to the it as autism

2.The paper presents a novel theory around the existence and potential role of innate randomness in decision making. The concept of innate randomness, however, is one that is proposed by the authors and there is no direct literature to support the theory as well as help with the interpretation of findings through that concept. This should be more clearly stated in the introduction as well as the discussion.

3.In Table 3, there are some randomness measures that appear to be statistically significant, however, the authors do not comment on those results in the Results and/or the Discussion sections of the paper. Could you please add a description and interpretation of these results or provide justification as to why these results are not part of the paper’s main findings.

4.The Figure legends need rewriting as they contain typos and they could also be a bit more descriptive.

5.Several sentences, especially in the Introduction, would benefit from rewriting. For example, in line 80 the following sentence “If all our choices need active consideration and thus involvement of system 2, this could impact our performance especially, concerning the time and effort spent in decision making (Croskerry & Nimmo, 2011; Moulton et al., 2007).” is difficult to follow. I would strongly advise the authors to re read the paper and amend those types of sentences, especially by being more thoughtful with the punctuation.

Experimental design

1.Could the authors please justify why they chose autism to study innate randomness? Why not choose OCD for example, where you would also expect more patterns in similar types of responses as well as difficulty in decision making.

2.As the task, is a modified version of the classic RNG, it would be really helpful if the authors could provide us with the task instructions to better understand what participants did differently, compared to the classic RNG task.

3.Could the authors please provide figures for all the randomness measures that they have examined using a windong approach?

Validity of the findings

1.Due to reasons that I have mentioned before and concern the fact that the authors present a new concept, that of innate randomness, the importance of replication of the study finding should be further highlighted. The authors do mention this in their discussion but I believe it is paramount to have this further strengthened.

2.Although the authors explain, in the Discussion, why the ASD group and the control group were successfully discriminated with machine learning, they offer no explanation and interpretation as to the set of variables that lead to this result and what they believe it could mean for innate randomness mechanisms and their function in ASD. Did they expect that these randomness variables would lead to such a result?

3.Are the paper’s results, when looking at randomness measures, in line with previous similar findings? Do the authors observe the same patterns of numbers and repetitions for example? If yes, then this could perhaps further support that the number sequences that they analyse were the product of innate randomness. This should be added in the discussion.

Cite this review as

·

Basic reporting

Dear Editor,
I really appreciate the opportunity to review the manuscript peerj-83452 entitled:
"Exploring randomness in autism"

Experimental design

I commend the authors for describing this critical and timely issue. The paper is interesting and well-written; however, I would like to highlight some issues that merit revision:

Validity of the findings

It is not particularly clear from the manuscript whether the assessed subjects had access to any form of psychotherapy or counseling, either in person or possibly remotely through telemedicine. Since this is a particularly important factor, e.g., cognitive behavioral therapy, its application could be a confounding factor. I would ask the authors to add a brief paragraph on this, including possibly listing it among the criteria for exclusion from recruitment for the study; if the data is not available it should be added to the limitations.

Cite this review as

---

## Round 0.2 · accepted · Accept

Many thanks for addressing all the issues.

Reviewer 1 ·

Basic reporting

Authors responded adequately to most of my comments and suggestions.
I find the manuscript, as it is now after their amendment, eligible for publication.

Experimental design

-

Validity of the findings

-

Additional comments

-

Cite this review as

·

Basic reporting

Dear Editor,
I really appreciate the opportunity to review the manuscript peerj-83452-R1 entitled:
"Exploring randomness in autism"

The paper is very interesting and well-written, methodologically unexceptionable, and the new implementations provide a valid contribution to the work. Every requested correction has been done, and the manuscript is now suitable for publication

Experimental design

No issues detected

Validity of the findings

No issues detected

Additional comments

No issues detected

Cite this review as